

# Genome-wide analysis of PRR gene family uncovers their roles in circadian rhythmic changes and response to drought stress in *Gossypium hirsutum* L.

Jingjing Wang[1,2,*], Zhaohai Du[1,*], Xuehan Huo[1,2], Juan Zhou[1], Yu Chen[1], Jingxia Zhang[1], Ao Pan[1], Xiaoyang Wang[3], Furong Wang[1,2] and Jun Zhang[1,2]

[1] Key Laboratory of Cotton Breeding and Cultivation in Huang-Huai-Hai Plain, Ministry of Agriculture and Rural Affairs, Cotton Research Center, Shandong Academy of Agricultural Sciences, Jinan, P. R. China
[2] College of Life Sciences, Shandong Normal University, Jinan, P. R. China
[3] State Key Laboratory of Cotton Biology, Institute of Cotton Research, Chinese Academy of Agricultural Sciences, Anyang, P. R. China
[*] These authors contributed equally to this work.

Corresponding authors
Furong Wang, wfr1125@126.com
Jun Zhang, zj0928@126.com

## ABSTRACT

**Background**. The circadian clock not only participates in regulating various stages of plant growth, development and metabolism, but confers plant environmental adaptability to stress such as drought. Pseudo-Response Regulators (PRRs) are important component of the central oscillator (the core of circadian clock) and play a significant role in plant photoperiod pathway. However, no systematical study about this gene family has been performed in cotton.

**Methods**. *PRR* genes were identified in diploid and tetraploid cotton using bioinformatics methods to investigate their homology, duplication and evolution relationship. Differential gene expression, KEGG enrichment analysis and qRT-PCR were conducted to analyze *PRR* gene expression patterns under diurnal changes and their response to drought stress.

**Results**. A total of 44 PRR family members were identified in four *Gossypium* species, with 16 in *G. hirsutum*, 10 in *G. raimondii*, and nine in *G. barbadense* as well as in *G. arboreum*. Phylogenetic analysis indicated that PRR proteins were divided into five subfamilies and whole genome duplication or segmental duplication contributed to the expansion of *Gossypium* PRR gene family. Gene structure analysis revealed that members in the same clade are similar, and multiple cis-elements related to light and drought stress response were enriched in the promoters of *GhPRR* genes. qRT-PCR results showed that *GhPRR* genes transcripts presented four expression peaks (6 h, 9 h, 12 h, 15 h) during 24 h and form obvious rhythmic expression trend. Transcriptome data with PEG treatment, along with qRT-PCR verification suggested that members of clade III (*GhPRR5a, b, d*) and clade V (*GhPRR3a* and *GhPRR3c*) may be involved in drought response. This study provides an insight into understanding the function of *PRR* genes in circadian rhythm and in response to drought stress in cotton.

## INTRODUCTION

The circadian clock is an autonomous endogenous biological rhythm that enables the living organisms to adapt to external daily and seasonal cycles, which play a significant role in plant growth and development for plant fitness (*Harmer, 2009*; *Hsu & Harmer, 2014*; *Lee et al., 2005*; *Mcclung, 2006*; *Uehara et al., 2019*). Although the circadian clock in different organisms is tissue-specific, most organisms have a conserved molecular mechanism-the core oscillator of positive and negative feedback loops formed at both the transcriptional and translational levels based on genome-wide gene expression regulation. (*Strayer et al., 2000*; *Harmer, 2009*; *Hsu & Harmer, 2014*; *Takata et al., 2009*; *Uehara et al., 2019*). Numerous studies have indicated that imperative roles for PRR gene family (PRR9, PRR7, PRR5, PRR3 and TOC1) in circadian clock (*Eriksson et al., 2003*; *Farre & Kay, 2007*; *Fujiwara et al., 2008*; *Gould et al., 2006*; *Ito et al., 2009*; *Kaczorowski & Quail, 2003*; *Nakamichi et al., 2020*; *Salome & McClung, 2005*; *Yamamoto et al., 2003*).

In *Arabidopsis thaliana*, the gene expression and protein expression levels of PRR family members have obvious circadian rhythmic expression pattern (*Matsushika et al., 2000*). PRR proteins contain two domains, the N-terminal contains a conserved PR (Pseudo reciever) domain, the C-terminus is a CCT domain, and CCT domain might interact with CONSTITUITIVE PHOTOMOR-PHOGENIC 1 (COP1) to control CONSTANS (CO) protein stability, and confer CO the ability to directly bind to DNA (*Makino et al., 2000*; *Jang et al., 2008*). PRRs could interact with CO at specific times and stabilize CO expression during the day, which promoting the CO protein to bind the promoter of *FLOWERING LOCUS T* (*FT*), inducing FT expression and promoting flowering (*Hayama et al., 2017*; *Kobayashi et al., 1999*; *Nakamichi et al., 2007*; *Song et al., 2012*). The CCT motif of PRRs is essential for recognizing key transcriptional factors such as *CCA1* (*CIRCADIAN CLOCK-ASSOCIATED 1*) and *LHY* (*LATE ELONGATED HYPOCOTYL*) to coordinate physiological processes with daily cycles (*Gendron et al., 2012*; *Kiba et al., 2007*; *Nakamichi et al., 2012*). Many studies showed that *PRRs* have role at circadian rhythmic expression levels in both transcriptional and protein levels, whether in continuous light or dark (*Más et al., 2003*; *Strayer et al., 2000*). Either in the *toc1* deletion mutant or *TOC1* overexpressing plants of *Arabidopsis thaliana*, the performance of the core oscillator has significant changes (*Huang et al., 2012*). Besides, PRR9, PRR7 and PRR5 could act as transcriptional repressors of CCA1 and LHY (*Nakamichi et al., 2010*).

At present, research mainly focuses on exploring the molecular mechanism of the photoperiod regulation pathway in *Arabidopsis thaliana*, and its regulation mechanism is becoming clear (*Song, Ito & Imaizumi, 2013*; *Wang, Kim & Somers, 2013*; *Wickland & Hanzawa, 2015*). Flowering time is an important factor affecting crop yield, thus dissection of photoperiod pathways regulating flowering time in crops and ornamental plants also becomes one of the hotspots in current researches (*Brambilla et al., 2017*; *Nakamichi, 2015*; *Yang et al., 2020*). However, molecular mechanisms of the photoperiodic control in crop flowering remain unclear. Only some studies on the cloning and functional analysis of PRR genes have been carried out in crops currently, such as rice (*Oryza sativa*) (*Murakami*

*et al., 2005*), wheat (*Triticum aestivum*) (*Nakahira et al., 1998*; *Beales et al., 2007*), barley (*Hordeum vulgare*) (*Turner et al., 2005*) and soybean (*Glycine max*) (*Liu et al., 2009*).

Flowering in an appropriate period has a critical effect on the fiber yield and quality of cotton, and there were only a few studies on genes related to flowering regulation in cotton (*Gossypium* spp.) (*Cai et al., 2017*; *Zhang et al., 2016*). With the completion of the genome sequencing of *Gossypium* species (*Du et al., 2018*; *Hu et al., 2019*; *Huang et al., 2020*; *Li et al., 2015*; *Wang et al., 2012a*; *Wang et al., 2012b*; *Wang et al., 2018*; *Yuan et al., 2015*; *Zhang et al., 2015*), the identification of new genes and the establishment of a new regulatory model would be helpful for studying the function of genes involved in cotton flowering pathways. Recently, a group also has reviewed a detailed study on other genetic bases of cotton drought tolerance (*Mahmood et al., 2020*).

In addition, the biological clock plays a vital role in adapting to external environmental stress, such as drought stress. In *Arabidopsis*, a triple mutant of prr9 prr7 prr5 confers drought stress tolerance by mediating cyclic expression of stress response genes, including DREB1/CBF (*dehydration-responsive element B1/C-repeat-binding factor*), which are regulated by the circadian clock (*Nakamichi et al., 2009*; *Fowler, Cook & Thomashow, 2005*). In soybeans, studies shown that drought stress affects the expression of circadian clock genes, and the expression of drought-responsive genes also has shown circadian rhythm (*Gome et al., 2014*). TOC1 has been shown to directly bind to the *ABAR* promoter region and regulate the periodic expression of *ABAR*, while ABA can up regulate TOC1. Therefore, TOC1 is considered to act as a molecular switch between the drought stress signaling pathway and the biological clock (*Legnaioli, Cuevas & Mas, 2009*)

Here, we identified 44 *PRR* genes from the four *Gossypium* species, and conducted basic bioinformatics analysis. We also investigated the expression pattern of *PRR* family members at the transcriptional level during 24 h. Further, we identified six *PRR* members responded to drought stress by analyzing transcriptome data with PEG treatment along with qRT-PCR verification. This study lays a foundation for studying the molecular mechanism of cotton photoperiod regulation and also provides an insight into understanding *PPRs* gene function in response to drought stress in cotton.

## MATERIALS & METHODS

### Identification of PRR gene family in *Gossypium* spp.

The domain numbered PF00072 (Response receiver domain) and PF06203 (CCT domain) in the Pfam database are often found in plant light signal transduction factors (*Sara et al., 2019*). Firstly, genome sequence of *G. hirsutum* (NAU-NBI v1.1 assembly genome), *G. arboretum* (CRI-updated_v1 assembly genome), *G. raimondii* (JGI_v2_a2.1 assembly genome) and *G. barbadense* (ZJU_v1.1 assembly genome) were downloaded from the Cottongen database (http://www.cottongen.org), respectively. This study used the protein sequences of 5 Arabidopsis PRRs were as queries to search the four *Gossypium* spp. proteomes through the basic local alignment search tool (BLAST, v 2.10.0) with default parameters ($E$-value $= 1 \times 10^3$) for each identified gene (*Altschul et al., 1990*). PR (Response receiver domain) and CCT domains, the typical PRRs domains,

were aligned and searched in HMMER 3.0 (https://www.ebi.ac.uk/Tools/hmmer/) (*Potter et al., 2018*). Next, sequences were searched and verified on the Conserved Domain Database (https://www.ncbi.nlm.nih.gov/Structure/cdd/wrpsb.cgi) and SMART (http://smart.embl-heidelberg.de/) (*Letunic et al., 2002*). Finally, the online site ExPASy Proteomics Server (http://www.expasy.org/) and Softberry (http://linux1.softberry.com/berry.phtml?topic=protcomppl&group=programs&subgroup=proloc) were used to analyze the physicochemical properties of the identified cotton PRR gene family, including amino acid number, nucleotide data, molecular weight, isoelectric point prediction and subcellular localization.

## Chromosomal locations, duplications, and synteny analysis of PRR gene members

Chromosomal location information for *PRR* genes was obtained from general feature format (gff) files of each cotton genomic databases and genes were mapped on the chromosomes using TBtools (*Chen et al., 2020*). Then MCScanX (*Wang et al., 2012a*; *Wang et al., 2012b*) was used to determine and analyze cotton *PRR* duplication and collinearity, Circos (http://circos.ca/) software were used to conducted image showing gene location and gene homology relationship.

## Phylogenetic analyses and gene structure organization of the PRR proteins in *Gossypium* spp.

To analyze evolutionary relationship, the PRR proteins sequence of various plant species including *Arabidopsis thaliana* (*Initiative, 2000*), Cocoa (*Theobroma cacao*) ((*Argout et al., 2011*)) and rice (*Oryza sativa*) (*Yu et al., 2005*) were downloaded from the *Arabidopsis* database TAIR10 (https://www.arabidopsis.org/), the plant genome database Phytozome 12 (http://phytozome.jgi.doe.gov/pz/portal.html) and EnsemblPlants (http://plants.ensembl.org/index.html), respectively. Multi-protein sequence alignment of the PRR proteins were aligned using MEGA7.0 (*Sudhir, Glen & Koichiro, 2016*), and constructed a phylogenetic tree using neighbor-joining (NJ) method with the bootstrap 1000. Finally, the evolutionary tree is visualized and beautified by the online software iTOL (https://itol.embl.de/) (*Letunic & Bork, 2019*). Location information of PRR members were obtained from gff files using SeqHunter1.0 (*Ye, Wang & Dou, 2010*) and the gene structures were displayed by the online software Gene Structure Display Server (GSDS 2.0) (http://gsds.gao-lab.org/) (*Guo et al., 2007*), and we performed motifs analysis on the online software MEME (http://meme-suite.org/) (*Bailey et al., 2009*) with following parameters: the maximum number discovered for the motif is 10, and the other parameters are default values. The graphic display is based on the Amazing optional gene viewer section in the software TBtools.

To identify the cis-elements in the promoter sequences of the 16 PRR family genes in *G. hirsutum*, the 2,000 bp of genomic sequences upstream of the start codon for each *PRR* gene were submitted to the online site PlantCARE (http://bioinformatics.psb.ugent.be/webtools/plantcare/html/), and the results are displayed by the Simple Bio Sequence Viewer in TBtools.

## Plant materials and treatment

The upland cotton (*G. hirsutum*) accession (Lumianyan 19, LMY 19) (*Li et al., 2004*), an early maturing variety, selected in this study were kept in our laboratory, planted in growth chamber (day/night temperature cycle of 28 °C light/25 °C dark with a 12-photoperiod), and samples were picked every 3 h from leaf in three-true-leaves stage. Germinated TM-1 cotton seeds were planted in the same photoperiod and temperature environment as LMY19, and treated with 400 mM polyethylene glycol (PEG6000) at the three-leaf stage from 7 am. TM-1 seedlings were divided into four treatment groups, treated with PEG for 0, 1, 3 and 6 h, respectively, and non-PEG treated seedling (treated with sterilized water) as control check at the same time point. Then we collected leaf samples at 0 h, 1 h, 3 h and 6 h after PEG treatment and non-PEG treated samples at each of the time point. Three biological replicates for each sample, the leaves from three seedlings as a biological replicate, and all samples were freezed with liquid nitrogen immediately and stored at −80 °C for qRT-PCR.

## RNA isolation and qRT-PCR analysis

The RNA was extracted from the samples using the Rapid Universal Plant RNA Extraction Kit (Huayueyang Biotechnology Co. Ltd.), and the Prime Scrip First Strand cDNA Synthesis Kit (Takara) used for reverse transcription, SYBR Premix Ex Taq II. (Takara) kit used for real-time PCR experiment, qRT-PCR analysis was carried out using SYBR Green on the Roche LightCycler® 480 II. The primers of PRR gene family were designed using Primer Premier 5.0 software and listed in Table S1, and the actin gene (AF059484) was selected as the internal reference gene (*Zhang et al., 2013*; *Zhang et al., 2013b*). The volume of the qRT-PCR reaction was 20 µL, and the amplification procedure was as follows: pre-denaturation at 95 °C for 30 s; denaturation at 95 °C for 5 s, annealing at 60 °C for 30 s, 40 cycles. Three biological and technical replicates were performed for the qRT-PCR tests. The relative gene expression levels were quantified by the $2^{-\Delta\Delta Ct}$ method (*Livak & Schmittgen, 2001*).

## Expression patterns and pathway enrichment analysis of PRR members

RNA-Seq data of *G. hirsutum* TM-1were obtained from the SRA database (PRJNA248163) (*Zhang et al., 2015*), and the FPKM (fragments per kilobase per million reads) values were calculated by RNA-seq data downloaded from the database of cottonFGD (*Zhu et al., 2017*).The gene expression pattern of *PRR* genes were displayed by R/pheatmap with the expression values normalized by $\log_2(FPKM+1)$. The expression profiles of all 16 *GhPRR* genes at different time of PEG treatment were further analyzed using R/Mfuzz. The differentially expressed genes (DEGs) were identified by DEseq2 (*Anders & Huber, 2010*). All detected genes in each sample were used to identify significantly DEGs (|log2 Foldchange|>1, $P < 0.05$) and KEGG analyses of DEGs were conducted in the Kyoto Encyclopedia of Genes and Genomes (KEGG) database for enrichment (*Kanehisa et al., 2014*), KEGG enrichment of DEGs was evaluated with KOBAS2.0 software (*Xie et al., 2011*) and bubble graph was displayed by R/ggplot2.

## RESULTS

### Genome-wide identification of PRR family genes in *Gossypium* spp

Based on multiple sequence alignment analysis, complete *PRR* genes were identified in four *Gossypium* species, including 16 in *G. hirsutum* (AD$_1$), 9 in *G. arboretum* (A$_2$), 10 in *G. raimondii* (D$_5$), and 9 in *G. barbadense* (AD$_2$). Additionally, we proceeded with *PRR* genes retrieved from plant genome database, with 5 in *Arabidopsis* (dicots), 5 in *rice* (monocots), and 6 in *cocoa* (dicot). All of them were renamed based on the homologous genes in *Arabidopsis* (Table S2). The number of *PRR* gene family in *G. hirsutum* (AtDt) was about twice as that in *G. arboreum* (A group) or *G. raimondii* (D group), it is consistent with the former one being tetraploid and the latter two being diploid. The basic information of *PRR* genes including protein sequence length, isoelectric points, and molecular weight in cotton were listed in Table S3 . The predicted GhPRR proteins ranged from 552 (GhPRR1a and GhPRR1b) to 795 (GhPRR3a) amino acids, with isoelectric points changed from 4.97 (GhPRR9a) to 8.42 (GhPRR3d) and molecular weight from 61.87 kDa (GhPRR1a) to 85.93 kDa (GhPRR3a).

### Chromosomal locations, duplications, and synteny analysis of PRR gene members

In order to display the chromosome distribution of *PRR* genes, mapping them on the corresponding chromosome. Eight of *GhPRR* genes were located on chromosomes of At sub-genome while five of *GhPRR* genes were on that of Dt sub-genome and three *GhPRR* genes were present in different scaffolds (Fig. S1). We further conducted whole genome collinearity analysis of 44 identified *PRR* genes in cotton, and explored the locus relationships between At and Dt sub-genomes as well as with A and D diploid cotton genomes (Fig. 1A, Table S4). There are 34 orthologous gene pairs were resulted from whole genome duplication or segmental duplication among *Gossypium* spp. Whole Genome duplication or segmental duplication was suggested to be the main causes of PRR gene family expansion in cotton (Table S5).

### Phylogenetic analyses and gene structure organization of the PRR proteins in *Gossypium* spp.

To investigate the evolutionary relationship of GhPRR proteins among mentioned seven species, phylogenetic tree was constructed (Fig. 1B). The PRR family of *Gossypium* was divided into five subgroups (clade I–V). There were 13 PRRs in Clade III (three GaPRRs, GbPRRs and GrPRRs respectively, four GhPRRs) and 11 PRRs (one GrPRR, two GaPRRs, four GbPRRs and GhPRRs individually) in clade IV. Clade I consisted of 9 PRRs (one GaPRR, two GbPRRs and GhPRRs singly, four GrPRRs), Clade V contained 7 PRRs (one GrPRR, two GaPRRs and four GhPRRs) and Clade II had 4 PRRs (one GaPRR and GrPRR respectively, two GhPRRs). GhPRRs were distributed throughout five subgroups (clade I–V), clade-I, clade-II and clade-IV containing PRRs from monocots and dicots simultaneously, illustrating that evolution of *GhPRR* genes in three clades occurred before the separation of monocots and dicots.

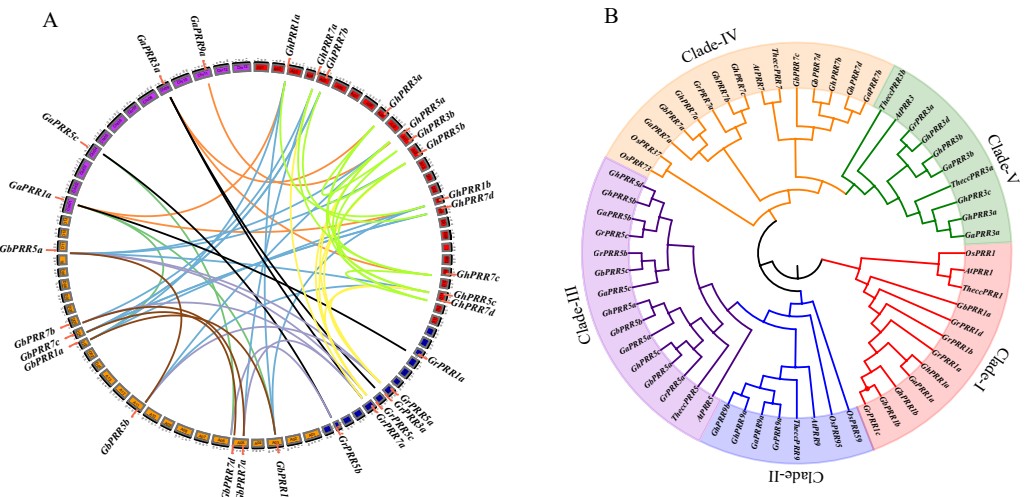

**Figure 1 Phylogenetic and collinearity analysis of PRR proteins in cotton.** (A) Gene duplication and collinearity analysis among cotton *PRR* genes (green lines and brown indicates paralogous genes in *G. hirsutum* and *G. barbadense*, orange lines indicates orthologous genes between *G. arboreum* and *G. hirsutum*, black lines indicates orthologous genes between *G. arboreum* and *G. raimondii*, blue lines indicates orthologous genes between *G. barbadense* and *G. hirsutum*, seagreen indicates orthologous genes between *G. arboreum* and *G. barbadense*, lightsteelblue indicates orthologous genes between *G. barbadense* and *G. raimondii*, yellow indicates orthologous genes between *G. raimondii* and *G. hirsutum*). Gene duplication and collinearity displayed on Circos (http://circos.ca/); (B) Phylogenetic tree of the *PRR* gene family.

PRRs protein in *G. hirsutum* was also divided into five subgroups (Fig. 2A), consistent with phylogenetic analyses. The motif distribution indicated that the order, size, and location of the motifs in the same subgroup were similar, but there were significant variety between different subgroups. Among them, 37.5% of the family members have the same sequence of motif structure: motif 4_9_3_1_7_5_6_10_8_2, while Clade-I contains the least number of motifs with only 5 motifs. All members of the PRR gene family contain motif1, motif2, motif3, motif4 and motif6, which are the conserved motifs of PRR family. In addition, the gene structure analysis exhibited that the distribution of introns and exons were similar among different subgroups, and the functional elements PR and CCT were distributed in both end side of each gene (Fig. 2B). All of member contained three PR structure elements, and most member contain two CCT domains, except that two members of the Clade-I subgroup contain one CCT domain.

To further analyze the transcriptional regulation and potential function of the *PRR* genes, the cis-elements in the promoter region were predicted (Fig. 2C). The results displayed that there are abundant regulatory elements existing in the promoter region, mainly focused on light response elements (G-Box, GT1-motif and TCT-motif, etc.), hormone responsive elements: abscisic acid response (ABRE), MeJA-response (CGTCA-motif and TGACG-motif), gibberellin-responsive element (TATC-box, P-box and GARE-motif), and stress responsive elements: drought-inducibility (MBS), low-temperature response (LTR), etc. There are 16, 14 and 6 *PRR* genes containing response elements to light, abscisic acid and drought stress, respectively. Motif sequences are often the binding sites of

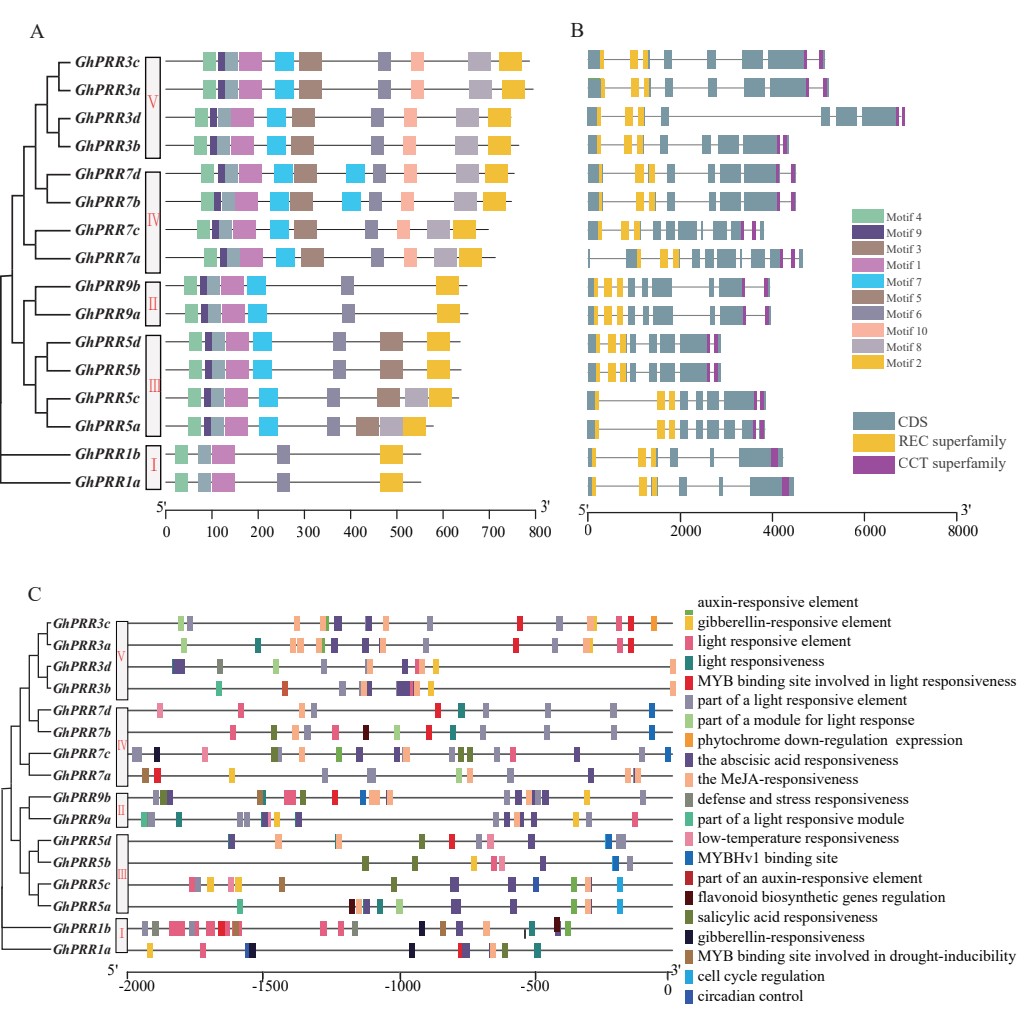

**Figure 2  Genetic structure and motif prediction of PRR members.** (A) Genetic structure of *GhPRR* genes; (B) motif prediction of GhPRR proteins; Length of each motif are shown proportionally. (C) Cis-elements prediction of *GhPRR* promoters. The scale bar is shown at the bottom.

some sequence-specific proteins (such as transcription factors), have important biological significance for important biological processes, such as RNA initiation, RNA termination, RNA cleavage, etc.

## The expression pattern of PRR members under diurnal changes

A feature shared by many clock gene transcripts is that their abundance is subject to diurnal oscillation. To analyze the peak transcripts of *GhPRR* s under diurnal cycle, the relative expression levels of *GhPRRs* together with its related genes (*GhFT* (*FLOWERING LOCUS T*), *GhCO* (*CONSTANS LIKE -2*), *GhLHY* (*LATE ELONGATED HYPOCOTYL*) and *GhCCA1* (*CIRCADIAN CLOCK-ASSOCIATED 1*)) during 24 h was detected by qRT-PCR (Fig. 3 and Table S6). The results showed that *GhLHY* -mRNA began to accumulate after dawn, and then mRNA of *GhPRR* genes began to reach the peak sequentially within a 24-hour period with multiple members at each peak. *GhFT*, *GhCO*, and *GhLHY* had

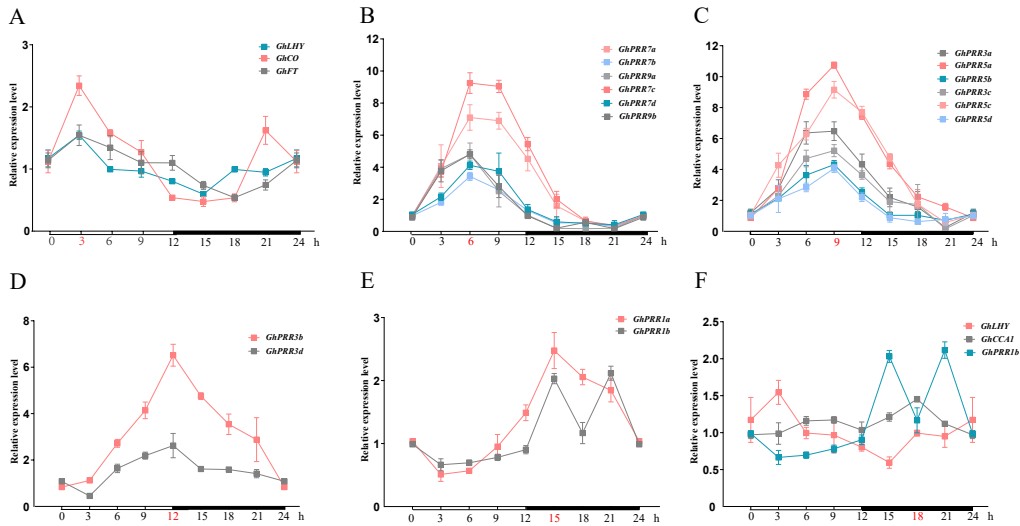

**Figure 3** **The expression pattern of PRR gene family and related genes during 24 h in LMY19.** White and black bars on X-axis indicate day and night conditions. Error bars represent means ± standard deviation ($n = 3$). (A) The expression pattern of GhLHY, GhCO and GhFT; (B) the expression pattern of GhPRR7a-d and GhPRR9a-b; (C) the expression pattern of GhPRR3a, GhPRR3c, and GhPRR5a-d; (D) the expression pattern of GhPRR3b and GhPRR3d; (E) the expression pattern of GhPRR1a and GhPRR1b; (F) the expression pattern of GhLHY, GhCCA1 and GhPRR1b.

the peak expression at 3 h after light. Subsequently, members inclade-II (*GhPRR9a* and *GhPRR9b*) and clade-IV (*GhPRR7a*, *GhPRR7b*, *GhPRR7c* and *GhPRR7d*) reached the expression peak after 6 h of light condition, and then members in clade III (*GhPRR5a*, *GhPRR5b*, *GhPRR5c* and *GhPRR5d*) and clade-V (*GhPRR3a* and *GhPRR3c*) at 9 h, another two members of clade-V (*GhPRR3b* and *GhPRR3d*) at 12 h. Finally, members (*GhPRR1a* and *GhPRR1b*) in clade-I reached expression peak after 3 h of dark. Additionally, the expression of *GhLHY* and *GhPRR1b* always showed an opposite trend during 24 h, it can be speculated that a mutual inhibition maybe exist between the two genes. These results indicated that expression of *GhPRR* genes has obvious rhythmic expression trend waves during 24 h.

## Identification of drought-stress related PRR genes in *G. hirsutum*

To investigate the roles for PRR genes in response to drought, we investigated the expression profile of *GhPRRs* under polyethylene glycol (PEG) treatment at 1, 3 and 6 h from the published transcriptome data sets. All detected genes in each sample were used to identify significantly DEGs (|log2 Foldchange|>1, $P < 0.05$) among PEG_1 h vs CK, PEG_3 h vs CK, PEG_6 h vs CK groups, and the PEG_6 h group contains the most number of DEGs (Table S7), so we selected the group data at 6 h treated with PEG for KEGG (Kyoto Encyclopedia of Genes and Genomes) analysis (Fig. 4A).The results revealed that the DEGs are mainly involved in circadian rhythm, photosynthesis, starch and sucrose metabolism, etc. (Fig. 4B). Six of *GhPRR* genes including three members in clade III (*GhPRR5a, b, d*) and two in clade-V (*GhPRR3a* and *GhPRR3c*) were involved in circadian rhythm pathway.

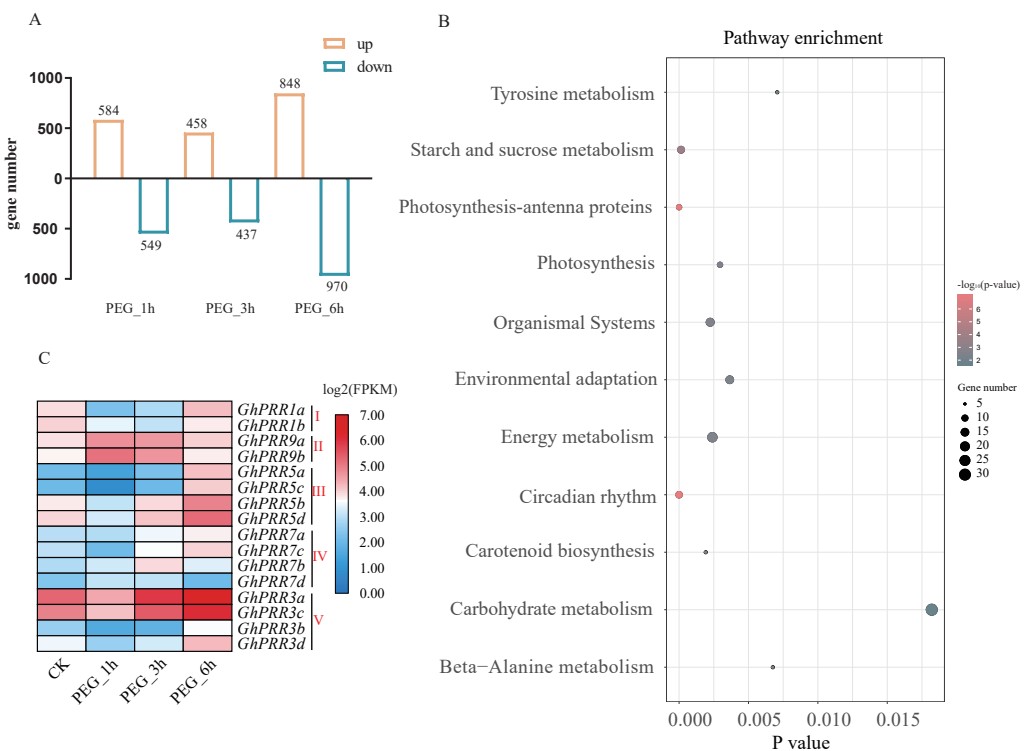

**Figure 4  Expression analysis and KEGG enrichment of *PRR* genes.** (A) Differential expression genes analysis; (B) KEGG pathway enrichment of DEGs PEG_6 h group; (C) expression pattern of *GhPRR* genes with PEG treated at CK, 1, 3 and 6 h. Count sizes of dots correspond to numbers of genes, and their colors correspond to -log10 (*p*-value) of pathway enrichment. DEGs: differentially expressed genes.

The expression patterns of these *GhPRR* genes have high expression level at 6 h with PEG treatment (Fig. 4C and Table S8), further analyzed and divided into 3 clusters, three members of clade III (*GhPRR5a-d*) and two of clade-V (*GhPRR3a* and *GhPRR3c*) in Cluster1 exhibited the same expression trend (Fig. S2 and Table S9), suggesting these *PRR* genes are significantly induced by PEG treatment. .

To further prove the expression changes of these genes at different time of PEG treatment (0 h, 1 h, 3 h and 6 h), the expression level of all member of PRR family were detected by qRT-PCR (Figs. 5A–5P and Table S10). The expression of genes (*GhPRR3a, c* and *GhPRR5a, b, d*) at the sixth hour after PEG6000 treatment was significantly higher than that of the blank control. All PRR genes displayed almost the similar expression changes compared with transcriptome data sets (CK, 1 h, 3 h, 6 h), and the correlation analysis between the transcriptome and qRT-PCR of *GhPRR* genes displayed by scatter plots, the result showed that the Pearson correlation coefficient log2 expression ratios calculated from qRT-PCR and RNA-seq of *GhPRR* genes was 0.78 (Fig. S3), suggesting the results are credible. It can be considered that genes mentioned above (*GhPRR3a, c* and *GhPRR5a, b, d*) maybe respond to drought stress.

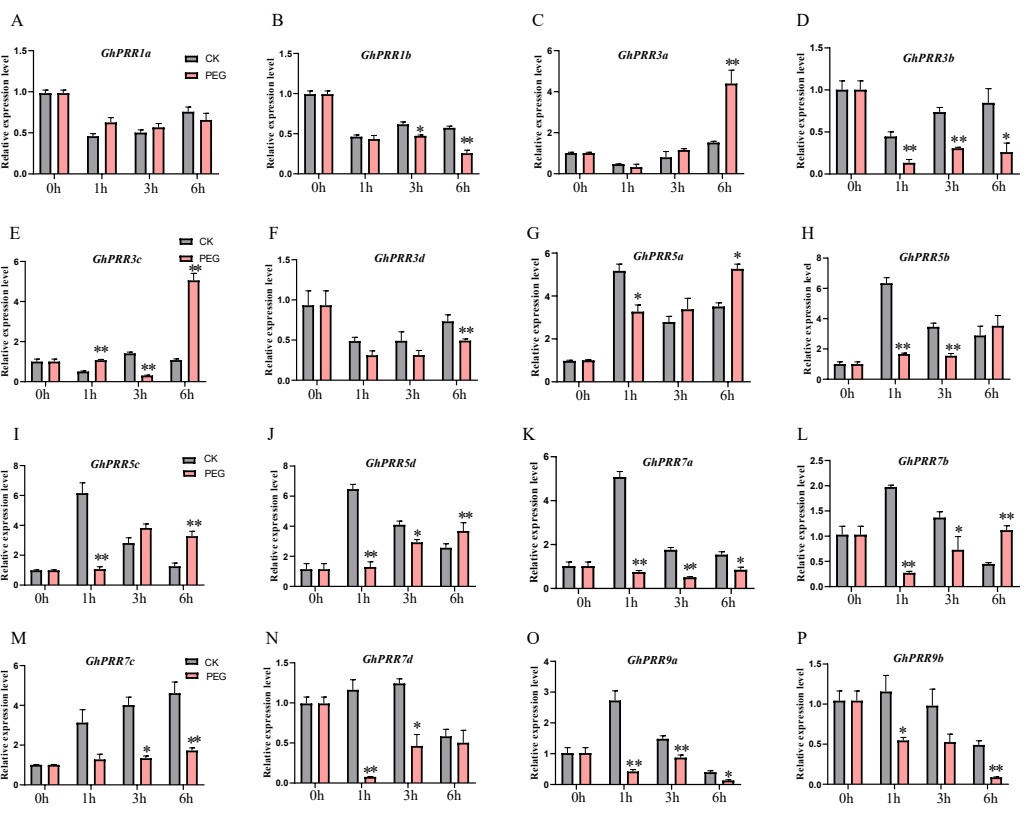

**Figure 5  qRT-PCR analysis of *PRR* genes under PEG treatment and non-PEG treatment (CK) at 0, 1, 3, 6 h.** The * and **indicate significant differences at $p < 0.05$ and $p < 0.01$ level, respectively. Differences analysis were compared using one-way ANOVA. (A–P) The relative expression of GhPRR1a,1b, 3a-d, 5a-d, 7a-d, 9a,9b under PEG treatment and non-PEG treatment (CK) at 0,1,3,6h, respectively.

## DISCUSSION

Light, one of the vital environmental factors, plays a significant role in promoting plant growth and development. Especially, with the alternating of sunrise and sunset, plants form a unique biological clock to regulate the growth and metabolic activities, like regulation of flowering time (*Hayama et al., 2017*; *Song et al., 2015*), hypocotyl elongation (*Seaton, Smith & Song, 2015*; *Soy et al., 2016*; *Zhu et al., 2016*), biotic (*Bhardwaj et al., 2011*; *Korneli, Danisman & Staiger, 2014*; *Zhang et al., 2013*; *Zhang et al., 2013b*) and abiotic stress response (*Keily et al., 2013*; *Nakamichi et al., 2009*), and so on.

Advances in cotton genomics and genetics recent years allowed us to perform a systematic study on *PRR* genes and to probe their potential functions in circadian clock. Here, sixteen *GhPRR* genes were identified totally in *G. hirsutum*, and phylogenetic tree were constructed to show the evolutionary relationship of PRR proteins in *G. hirsutum* and other plant species (Fig. 1B). The PRR family of *Gossypium* was divided into five subgroups (Clade I–V), which consistent that of in *Arabidopsis* (PRR1 (TOC1), PRR3, PRR5, PRR7, PRR9). Orthologue genes always share identical biological functions over evolutionary stages (*Altenhoff & Dessimoz, 2009*), the exon-intron structure and the motif distribution of

*GhPRR* genes in the same subgroup were similar. According to chromosomal localization and genomic collinearity analysis, it can be speculated that due to the hybridization of A and D subgenome in the *G. hirsutum*, the gene amplification is carried out by tandem repeat and fragment replication (*Jackson & Chen, 2010*). There is a high degree of collinearity between the *PRR* genes of the At and the Dt subgenome of the tetraploid *G. hirsutum* (*Li et al., 2015*). In this study, 14 (7 pairs) of 16 PRR members are orthologous genes, indicating that *G. hirsutum* has undergone large-scale gene rearrangement at the genomic level during species formation, which is consistent with the results of the allotetraploid *G. hirsutum* genome (*Wang et al., 2018*; *Li et al., 2015*; *Zhang et al., 2015*).

A large number of experimental studies have been carried out about circadian clock in *Arabidopsis* (*Alabadí et al., 2001*; *Más et al., 2003*; *Legnaioli, Cuevas & Mas, 2009*). PRRs proteins interact with CCA1 and LHY through complex mechanisms, playing a vital role in the growth and development, flowering induction and metabolic regulation of plants (*Harmer, 2009*; *Legnaioli, Cuevas & Mas, 2009*; *Mizuno & Nakamichi, 2005*). The function of some circadian clock-related genes has been cloned and verified based on gene homology in major crops, such as rice, soybean (*Gome et al., 2014*; *Xue et al., 2012*; *Yang et al., 2013*). So far, circadian clock regulation mechanism in cotton is still a mystery, only one study has identified *Gh_D03G0885* (*GhPRR1b*) as a candidate gene for cotton early maturity traits using genotyping-by-sequencing (*Li et al., 2017*). TOC1 (known as Pseudo Response Regulator, PRR1) is an important component of the core oscillator and closed positive and negative feedback loop with LHY (Late Elongated Hypocotyl) and CCA1 (Circadian Clock Associated 1), formulating the basic framework of the *Arabidopsis* circadian clock core oscillator (*Alabadí et al., 2001*; *Gendron et al., 2012*; *Huang et al., 2012*).

Further, qRT-PCR analysis revealed that the relative expression of PRR members had apparent rhythmic expression trend among 24 h, which similar with that of PRR members *(PRR1/TOC1, PRR3, PRR5, PRR7, PRR9)* in *Arabidopsis*. Transcript expression peaks appear in the order of *PRR9*, *PRR7*, *PRR5*, *PRR3* and *TOC1* (*PRR1*) in *Arabidopsis* (*Matsushika et al., 2000*), while four expression peaks appeared in this study and there were multiple members at each peak, speculating that it is related to chromosome doubling in the process of forming allotetraploid in *G. hirsutum* (*Jackson & Chen, 2010*). The *PRR1a* gene had the last peak of expression and highly expressed at night, which consistent with that of *APRR1* in *Arabidopsis* (*De Caluwé et al., 2016*), while *GhPRR1b* has two peak of expression at night in this study. Therefore, detailed study should be carried out about this the gene in cotton.

In cotton, *GhLHY*-mRNA began to accumulate after dawn, and then members of the GhPRR gene family began to reach the peak sequentially within a 24-hour period, which is consistent with the results in *Arabidopsis*. The *GhPRR1b* gene has high homology with *PRR1* in *Arabidopsis thaliana* by alignment, so it is speculated that *GhPRR1b* is the core component of the circadian clock in *G. hirsutum*. *GhPRR1b* and *GhLHY* have opposite expression trends among 24 h, and there may be a mutual inhibition between *GhPRR1* and *GhLHY,* the expression trends of which consistent with *PRR1* gene in *Arabidopsis thaliana*. As an inhibitor of circadian clock gene expression, TOC1 gene can inhibit the expression of most circadian clock core genes, and affect flowering pathway of photoperiod regulation

by controlling the function of circadian clock (*Strayer et al., 2000*; *Pokhilko et al., 2012*). As an important factor in the export pathway of the circadian clock, CO protein has been proved in *Arabidopsis* to confirm the stability of PRRs protein-mediated CO expression, and can enhance the binding of CO to *FT* promoter, then *FT* start transcribe and promote flowering (*Jang et al., 2008*). The pathway of PRRs family members mediate the stability of CO expression still needs further experiments in cotton.

In addition, there are many studies focus on the response of circadian clock to abiotic stress in crops (*Flowers, 2004*; *Lu et al., 2017*; *Zhang et al., 2020*). TOC1 can bind to the *ABAR* promoter of ABA-related genes and regulate its circadian rhythm expression, and can be thought to act as molecular switches between drought stress signaling pathways and circadian clocks in *Arabidopsis* (*Legnaioli, Cuevas & Mas, 2009*). In soybeans, studies have also shown that drought stress affects the expression of circadian clock genes, and the expression of drought-responsive genes also has circadian rhythm (*Gome et al., 2014*). Based on these researches, this study identified 16 PRR members in cotton and analyzed the expression pattern of PRR genes during 24 h and in response to drought stress. The result showed that PRR members expression display obvious rhythmic expression trend and six of them may be involved in responding to drought stress, which is helpful to understand the evolution and function of the PRRs gene family, and provide thoughts and clues for further study the function of the PRR gene family in cotton.

## CONCLUSIONS

In this study, we identified 44 *PRR* genes in cotton (*Gossypium* spp.) and classified them into five subgroups based on the phylogenetic tree. Then we systematically analyzed PRRs in cotton (*Gossypium* spp.), including the domains, the gene structure, promoter cis-acting element, chromosome localization distribution and collinearity analysis. In addition, we also investigated the evolutionary relationship of PRRs among *G. hirsutum*, *G. barbadense*, *G. arboreum* and *G. raimondii*, *Arabidopsis thaliana*, *Theobroma cacao* and *Oryza sativa*. Moreover, qRT-PCR results showed that the expression of members of PRRs family has obvious rhythmic expression trend, and gene differential expression and KEGG enrichment analysis of the transcriptome data with PEG treatment, along with qRT-PCR verification altogether demonstrated members of clade III (*GhPRR5a, b, d*) and two members of clade-V (*GhPRR3a* and *GhPRR3c*) are significantly induced by PEG treatment, so it is speculated that these *GhPRR* genes may be involved in drought response. This study will provide a theoretical basis for studying the function of *PRRs* in cotton.

## ACKNOWLEDGEMENTS

The authors thank Guoyong Fu and Tahir Mahmood for kindly revising this manuscript.

### Funding

This work is supported by the National Science Foundation in China (31671742); the National Project of Modern Agricultural Industry Technology System in China [CARS-15-05]; the Taishan Scholars Program of Shandong Province [ts201511070]. The funders had no role in study design, data collection and analysis, decision to publish, or preparation of the manuscript.

### Grant Disclosures

The following grant information was disclosed by the authors:
National Science Foundation in China: 31671742.
National Project of Modern Agricultural Industry Technology System in China: CARS-15-05.
Taishan Scholars Program of Shandong Province: ts201511070.

### Competing Interests

The authors declare there are no competing interests.

### Author Contributions

- Jingjing Wang conceived and designed the experiments, performed the experiments, prepared figures and/or tables, authored or reviewed drafts of the paper, and approved the final draft.
- Zhaohai Du, Xuehan Huo and Juan Zhou performed the experiments, prepared figures and/or tables, and approved the final draft.
- Yu Chen, Jingxia Zhang, Ao Pan and Xiaoyang Wang analyzed the data, prepared figures and/or tables, and approved the final draft.
- Furong Wang and Jun Zhang conceived and designed the experiments, authored or reviewed drafts of the paper, and approved the final draft.

### Data Availability

The raw data of qPCR is available in Tables S6 and Table S10, and the "Results". The RNA sequence raw data is available at the SRA database: PRJNA490626.

### Supplemental Information

Supplemental information for this article can be found online at http://dx.doi.org/10.7717/peerj.9936#supplemental-information.

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
