# Peer review of "Genome-wide analysis of PRR gene family uncovers their roles in circadian rhythmic changes and response to drought stress in Gossypium hirsutum L"

_PeerJ, doi:10.7717/peerj.9936_

## Round 0.1 · original submission · Major Revisions

Please take into consideration the reviewer's comments, and provide a revised manuscript and a detailed point-by-point rebuttal letter.

Reviewer 1 ·

Basic reporting

The authors analyzed cotton PRR genes, regarding to circadian rhythm and drought stress responses. This is the first report of PRR genes in cotton. Although I have comments on the manuscript, I feel this paper add significance of importance of PRR family in circadian clock in cotton.

Major comment
Gene expression analysis in Figure 5.
Proper controls of PEG treatment are required. In other word, the experiment should contain non-PEG treated samples at 1h, 3h, and 6h. I could not distinguish whether PEG treatment or time-of-day affect PRR genes expression, in this layout. For example, GhPRR3c was unregulated at PEG 6h, suggesting that either PEG treatment, the clock, or both induce GhPRR3c.

Minor comments
Line 49. I recommend to add a paper describing TOC1 cloning as reference (DOI: 10.1126/science.289.5480.768).

Line 54. The N-terminal of PRR is PR (Pseudo reciever) domain, not REC (Response regulator reciever) (DOI: 10.1093/pcp/41.6.791). REC is involved in the His-Asp phosphorelay signal transduction, whereas PR is not (DOI: 10.1093/pcp/41.6.791). Instead, PR domain of PRR family is crucial for protein-protein interactions (doi: 10.1074/jbc.M803471200, doi: 10.1105/tpc.107.053033). It is possible to find PRR proteins using homology search using REC and CTT; however, precise description of domain name of PRR is essential to avoid misinterpretation by readers.

Line 59. In addition to binding to CO, CCT motif of PRR proteins are required for association with target gene promoter such as CCA1 and LHY (doi: 10.1073/pnas.1205156109, DOI: 10.1073/pnas.1200355109). The PRRs repress expression of CDF genes that encode transcriptional repressor of CO and FT (doi: 10.1073/pnas.1205156109, DOI: 10.1126/science.1219644).

Line 71. PRR genes function in wheat was reported by other workers (doi: 10.1007/s00122-007-0603-4).

Line 83. In addition to TOC1, triple mutants of prr9 prr7 prr5 confers drought stress tolerance by upregulation of DREB1 (CBF) genes in Arabidopsis (doi: 10.1093/pcp/pcp004). Arabidopsis DREB1 genes are regulated by the circadian clock (doi: 10.1104/pp.104.058354).

Please cite proper papers in addition to above recommendation. For example, Farre and Kay 2005 paper found in the introduction is not cited in reference section.

Experimental design

Gene expression analysis in Figure 5.
Proper controls of PEG treatment are required. In other word, the experiment should contain non-PEG treated samples at 1h, 3h, and 6h. I could not distinguish whether PEG treatment or time-of-day affect PRR genes expression, in this layout. For example, GhPRR3c was unregulated at PEG 6h, suggesting that either PEG treatment, the clock, or both induce GhPRR3c.

Validity of the findings

I have no comment.

Additional comments

The authors analyzed cotton PRR genes, regarding to circadian rhythm and drought stress responses. This is the first report of PRR genes in cotton. Although I have comments on the manuscript, I feel this paper add significance of importance of PRR family in circadian clock in cotton.

Major comment
Gene expression analysis in Figure 5.
Proper controls of PEG treatment are required. In other word, the experiment should contain non-PEG treated samples at 1h, 3h, and 6h. I could not distinguish whether PEG treatment or time-of-day affect PRR genes expression, in this layout. For example, GhPRR3c was unregulated at PEG 6h, suggesting that either PEG treatment, the clock, or both induce GhPRR3c.

Minor comments
Line 49. I recommend to add a paper describing TOC1 cloning as reference (DOI: 10.1126/science.289.5480.768).

Line 54. The N-terminal of PRR is PR (Pseudo reciever) domain, not REC (Response regulator reciever) (DOI: 10.1093/pcp/41.6.791). REC is involved in the His-Asp phosphorelay signal transduction, whereas PR is not (DOI: 10.1093/pcp/41.6.791). Instead, PR domain of PRR family is crucial for protein-protein interactions (doi: 10.1074/jbc.M803471200, doi: 10.1105/tpc.107.053033). It is possible to find PRR proteins using homology search using REC and CTT; however, precise description of domain name of PRR is essential to avoid misinterpretation by readers.

Line 59. In addition to binding to CO, CCT motif of PRR proteins are required for association with target gene promoter such as CCA1 and LHY (doi: 10.1073/pnas.1205156109, DOI: 10.1073/pnas.1200355109). The PRRs repress expression of CDF genes that encode transcriptional repressor of CO and FT (doi: 10.1073/pnas.1205156109, DOI: 10.1126/science.1219644).

Line 71. PRR genes function in wheat was reported by other workers (doi: 10.1007/s00122-007-0603-4).

Line 83. In addition to TOC1, triple mutants of prr9 prr7 prr5 confers drought stress tolerance by upregulation of DREB1 (CBF) genes in Arabidopsis (doi: 10.1093/pcp/pcp004). Arabidopsis DREB1 genes are regulated by the circadian clock (doi: 10.1104/pp.104.058354).

Please cite proper papers in addition to above recommendation. For example, Farre and Kay 2005 paper found in the introduction is not cited in reference section.

Reviewer 2 ·

Basic reporting

The introduction is well supported. The materials and methods, and figures legends are not properly described so it is difficult to understand the information presented.

For a better understanding of the data presented, the authors should improve the legends of the figures. These must contain all the necessary to understand the information presented. In addition, it is recommended to improve the text quality used in the figures.

Figure 1: Check the letters of the panels. The phylogenetic tree is represented by letter B but on the legend is indicated as A.

Figure 2: In panel A, for a better understanding of the information presented, it is recommended that authors indicate which figures correspond to each analysis, ex: Genetic structure (right side). In panel B, for a better understanding of the figure, I suggest indicating the X's axis the coordinates of the regulatory elements respect to the transcription start site (ex -50). In addition, the color assigned to flavonoid biosynthetic genes regulation is missing.

Figure 3: To understand the results obtained, it is recommended to include more information, including What mean the black and white bars showed on X-axis? What represent the error bars showed?

Figure 4: It is necessary to indicate the values used as filters to consider the differentially expressed genes (p-value, FDR, fold change, etc). What means "count"?

Experimental design

The information shown in the materials and methods is incomplete and not clear, it is necessary that this part be substantially improved to support the results showed.

It is unclear how they identified the PRR candidate genes. The strategy shown is confusing and more details need to be included. I suggest checking the wording, indicate the databases used, and add the e-value used as cut-off on the blast analysis.

About biological replicates, more information on experimental design should be included, how many plants for each experiment were used? How many plants by biological replicate were used?

It is necessary that the authors describe in more detail Why LMY19 and TM-1 varieties/accessions were used.

150-157: It is necessary to check the expression patterns and pathway enrichment analysis of PRR members. The reference showed (Zhang et al., 2015) does not correspond to the BioProject (PRJNA490626) and SRA accession numbers shown. These accession numbers correspond to the work reported by Hu et al., 2019 (doi: 10.1038 / s41588-019-0371-5), and they were registered on September 13, 2018, in the NCBI (https: //www.ncbi.nlm. nih.gov/bioproject/?term=PRJNA490626). Additionally, to understand how these data are related to the work reported here, it is necessary to include more information about the transcriptome data set used, including the description of the experiment, description of each sample (treatments and control), etc. In this sense, the SRA accession number of the control sample is missing.

To support the results obtained, the authors should describe in more detail the methodology used for the analysis of the transcriptome data set, including filtering of reads by quality, mapping to the reference, quantification of the expression levels, the filters used to define differentially expressed genes (p-value, FDR, fold change), etc. To enrichment analysis of KEGG, which data set was used as reference to identify the pathways enriched?

Please, explain why do you use the "mpoly" to identify differentially expressed genes instead of DEGSeq2, which was used by Hu et al., 2019 (where the data used comes from).

Validity of the findings

It is necessary to show the homology evidence of PRR sequences in the comparison with Arabidopsis, I suggest including the E-value, identity percentage, and query coverage percentage obtained for each comparison showed on Table S2.

The authors performed a search of motifs in the PRR sequences, they have found clear differences between the sequences, it is necessary to describe the possible biological importance of the presence or absence of these motifs.

Regarding the search for regulatory elements in the promoters of PRR genes, it is necessary to present the analysis and discussion of how many genes have response elements to light, abscisic acid, and drought stress.

In the expression patterns analysis of PRR genes under diurnal change, the authors claim that "the GhPRR genes have obvious circadian waves during 24 hours", however, the results as presented do not clearly support this conclusion. Some ideas are not clear, for example, the authors claim that “GhPRR genes reached the peak every 3 h after 6 h of light condition…” However, as figure 3 showed, it seems that the GhPRR7a, 7b, 9a, 7c, 7d, and 9b genes presented the maximum peak after 6h of darkness. While, the GhPRR3a, 5a, 5b, 3c, 5c, and 5d genes the maximum peak is observed after 9h of darkness. Please, verify the information showed on the figure and explain this part in more detail. It is necessary to improve de legend of this figure.

The authors only show the enrichment analysis of the KEGG pathways for the 6h treatment, please, explain why the other times were not analyzed. It is recommended to show the complete analysis of PEG treatment, for both induced and repressed genes.

The authors mention that "... genes under PEG treatment showed that these six members have the same expression trend and high expression level at 6 h ..." (228-231). To support this idea, it is necessary to carry-out a hierarchical clustering of genes based on their expression patterns, this will help to confirm if these genes have similar expression patterns.

233-234: The authors claim that "All PRR genes displayed almost the similar expression changes compared with transcriptome data sets". To support this, it is necessary to show a graph with the gene expression levels obtained in the transcriptome vs qRT-PCR analyzes or the correlation value between both analyzes.

258-275: In this part, the authors give a good description of the reports on the circadian clock in plants and the role of the PRR proteins in this molecular mechanism, however, it is not completely clear what is the contribution of the results reported here to this area. It is necessary that the results obtained must be compared and discussed in greater depth.

Additional comments

93-94 the idea is not clear. Please, explain it. I suggest checking the wording.

102-103 the idea is not clear, what the authors mean with “Finally, the whole genome sequence of Arabidopsis thaliana (Initiative et al., 2000), Cocoa (Theobroma cacao) (Argout et al., 2011) and rice (Oryza sativa) (Yu et al., 2005) were loaded from the Arabidopsis database TAIR10 (https://www.arabidopsis.org/), the plant genome database Phytozome 12 (http://phytozome.jgi.doe.gov/pz /portal.html) and EnsemblPlants (http://plants.ensembl.org/index.html), respectively ” Please, explain it. I suggest change the word "loaded" by "download".

Check the use of the word “respectively”, in some cases it is unnecessary, for example, line 136. It should say "after treatment".

It is necessary to homogenize the writing style, the difference between the introduction and materials and methods is remarkable.

136: It is necessary to include the polymerization degree of the PEG used.

164-166: In this part, it is not clear which genome is tetraploid and which diploids. Please, check it.

168: The amino acid and isoelectric point numbers showed in the text do not correspond to those indicated in table S3. Please, check it.

209: For a better understanding, I suggest indicating the names of the products of the GhFT, GhCO, GhLHY, and GhCCA1 genes.

285-286: It is not clear what the authors mean with "... expression of GhPRR follows the expression of GhLHY". Please, explain it.

288: Arabidopsis thaliana should be in italics.

---

## Round 0.2 · Minor Revisions

Please take into consideration the reviewer’s comments and send back a revised version addressing those concerns.

Reviewer 1 ·

Basic reporting

The revised manuscript was improved. Especially, the authors added new experiment to dissect drought and clock effects on PRR genes expression.
I appreciate the authors’s effort, but I have still comments as below.

1. Jang et al., EMBO J 2008 nor Makino et al., PCP 2000, did not show function of the CCT domain of PRR proteins. The CCT domain of PRR proteins were studied well by Gendron et al., PNAS 2012 and Nakamichi et al., PNAS 2012 as the authors wrote.

2. ‘Periodic expression’, ’rhythmic expression’, ‘cyclic expression’, ‘circadian’ and ‘oscillation’ are generally used if researchers analyzed samples at least 2-days and found 2-peaks or 2-troughs of interested biological activities. The authors analyzed expression of genes in 1 day (Figure 3), I recommend that these words are not used to explain expression patterns of the genes (‘periodic’ in Line 429, ‘circadian rhythm’ in Line 439, ‘circadian waves’ in Line 449).

3. It was unclear how the authors choose two samples to get DEG in Figure 4A. Did the authors compare transcriptome of PEG_6h and that of untreated_0h to identify DEG in PEG_6h columns? I also request the information when (time-of-day) the authors started to treat plants with PEG.

4. While the authors wrote that ‘The expression of the above mentioned genes (GhPRR5a-d, GhPRR3a and GhPRR3c) at the sixth hour after PEG6000 treatment was significantly higher than that of the blank control’, I found that GhPRR5b expression at 6h-CK and 6h-PEG are similar.

Experimental design

It was unclear how the authors choose two samples to get DEG in Figure 4A. Did the authors compare transcriptome of PEG_6h and that of untreated_0h to identify DEG in PEG_6h columns? I also request the information when (time-of-day) the authors started to treat plants with PEG.

Validity of the findings

well documented.

---

## Round 0.3 · accepted · Accept

Thanks for addressing the minor revisions requested. Now your manuscript is accepted in PeerJ.